# Virus-Like Particles Produced Using the Brome Mosaic Virus Recombinant Capsid Protein Expressed in a Bacterial System

**DOI:** 10.3390/ijms22063098

**Published:** 2021-03-18

**Authors:** Aleksander Strugała, Jakub Jagielski, Karol Kamel, Grzegorz Nowaczyk, Marcin Radom, Marek Figlerowicz, Anna Urbanowicz

**Affiliations:** 1Institute of Bioorganic Chemistry, Polish Academy of Sciences, 61-704 Poznan, Poland; astrugala@ibch.poznan.pl (A.S.); kamel@ibch.poznan.pl (K.K.); Marcin.Radom@cs.put.poznan.pl (M.R.); Marek.Figlerowicz@ibch.poznan.pl (M.F.); 2NanoBioMedical Centre, Adam Mickiewicz University, 61-614 Poznan, Poland; jakub.jagielski@amu.edu.pl (J.J.); nowag@amu.edu.pl (G.N.); 3Institute of Computing Science, Poznan University of Technology, 60-965 Poznan, Poland

**Keywords:** virus-like particles, brome mosaic virus, capsid, self-assembly

## Abstract

Virus-like particles (VLPs), due to their nanoscale dimensions, presence of interior cavities, self-organization abilities and responsiveness to environmental changes, are of interest in the field of nanotechnology. Nevertheless, comprehensive knowledge of VLP self-assembly principles is incomplete. VLP formation is governed by two types of interactions: protein–cargo and protein–protein. These interactions can be modulated by the physicochemical properties of the surroundings. Here, we used brome mosaic virus (BMV) capsid protein produced in an *E. coli* expression system to study the impact of ionic strength, pH and encapsulated cargo on the assembly of VLPs and their features. We showed that empty VLP assembly strongly depends on pH whereas ionic strength of the buffer plays secondary but significant role. Comparison of VLPs containing tRNA and polystyrene sulfonic acid (PSS) revealed that the structured tRNA profoundly increases VLPs stability. We also designed and produced mutated BMV capsid proteins that formed VLPs showing altered diameters and stability compared to VLPs composed of unmodified proteins. We also observed that VLPs containing unstructured polyelectrolyte (PSS) adopt compact but not necessarily more stable structures. Thus, our methodology of VLP production allows for obtaining different VLP variants and their adjustment to the incorporated cargo.

## 1. Introduction

Viral capsids are natural nanostructures that evolved towards the efficient delivery of viral genetic material to host cells. Their nanoscale dimensions, presence of an interior cavity, self-organization abilities and responsiveness to environmental conditions are desired features in terms of nanotechnology. Additional attributes of viral capsids, including homogeneity, susceptibility to chemical modification and, in the case of plant viruses, lack of pathogenic activity to humans, give them potential as biologically active compound carriers in the biomedical field. The reports on potential applications of viral capsids in nanotechnology and biomedical fields has been thoroughly reviewed [1,2,3,4]. The key to the utilization of viral capsids is understanding and controlling their assembly and disassembly processes.

Brome mosaic virus (BMV) is one of the most extensively studied plant viruses. Its structure and biology are well known, and numerous reports describe the potential applications of BMV as a vehicle for nanoparticle delivery. BMV is a typical representative plant virus with a genome composed of single-stranded RNA and nonenveloped spherical virions. BMV belongs to the Bromovirus genus. The BMV capsid is composed of 180 capsid protein (CP) subunits with a weight of 20 kDa. The virion has an approximately 28 nm diameter, depending on the surrounding conditions, and it shows T = 3 icosahedral symmetry [5]. The cargo space inside the virion is limited by the internal cage with the T = 1 triangulation number. CP consists of 189 amino acids and has a globular shape with a protruding N-terminal arginine-rich motif (ARM) directed into the interior of the capsid. The ARM, due to its positive charge, interacts with negatively charged RNA and therefore is crucial for capsid assembly.

Capsid assembly is currently being extensively analyzed, and some general rules of this process have already been determined. The formation of capsid depends on the temperature, pH, ionic strength, CP concentration and features of the encapsidated core molecules [6,7,8]. Due to the recent development of experimental and computational techniques in the field of physical virology, the static descriptions of viral particles have been replaced by dynamic models based on soft-mode dynamics and kinetic trapping [9]. Still, the knowledge of the capsid assembly is not sufficient to control it or rationally design VLPs with desired features. The disassembly and reassembly of BMV virions containing their native RNA or RNA derived from another virus is possible in the course of dialysis, starting from a high-salt to a low-salt buffer at neutral pH [10,11,12]. Under natural conditions, BMV capsids as well as those of similar viruses, such as cowpea chlorotic mottle virus (CCMV), begin forming via the nucleation process, driven by electrostatic interactions between CP dimers and the viral genome. The assembly of complete virions is further continued mostly due to interactions among CPs [13]. The key to BMV virion assembly is the approximately 200-nucleotide-long 3′ end of the genomic RNA, which forms a tRNA-like structure (TLS). The importance of TLSs was initially shown by Choi et al. in an experiment in which BMV capsid assembly attempted upon RNA with truncated TLSs failed, while the addition of yeast tRNA promoted virion formation [14]. Later, it was shown that not only RNA but also other materials, such as gold nanoparticles, quantum dots [15], and even cubic-shaped nanoparticles [16], may be encapsulated in the BMV capsid. The primary requirement for in vitro encapsidation of any type of cargo is the negative charge forming electrostatic interactions with the positively charged N-terminal portion of the BMV CP. Under in vitro conditions, these interactions are heavily dependent on the ionic strength of the buffer and as such can be modulated [6,13,17].

The capsid assembly process is driven not only by CP-cargo interactions but also by interactions among the CPs. The importance of such interactions can clearly be seen in the formation of CP dimers, which are the basic building blocks for capsid structures. CP dimers attach to each other, creating larger protein complexes and leading to full capsid formation [7]. The interactions among CPs are based on hydrophobic attractions and hydrophilic repulsions, in contrast to the interactions between CP and cargo [13]. Assembly of BMV-derived empty capsids is pH-dependent and, according to molecular dynamics and dynamic light scattering (DLS) analysis, is initiated by the formation of CP trimer nuclei at pH close to 5 [18].

Considering the utilization of plant viral capsids as biologically active compounds vehicles, attempts are made to obtain VLPs showing desired features. The easiest way to modify VLPs is to introduce mutations into the CP sequence. However, during viral replication in native host plants, most CP mutations result in the loss of viral infectivity or are reversed to the wild-type sequence [19]. This phenomenon is related to error-prone RNA replication, RNA recombination and subsequent selective pressure, which eliminates viral variants with lower fitness compared to the wild type [19,20,21,22]. To overcome this impediment, heterologous expression systems are applied that offer CP production without selective pressure working on protein mutants.

In this paper, we describe the in vitro assembly of VLPs with the application of recombinant BMV CP (rCP) produced in a bacterial expression system. We used the rCP to study the impact of environmental factors and encapsulated cargo on the assembly process and properties of the resulting VLPs. In our assay, we applied a spectrum of methods, including protein crosslinking, electrophoresis, microscale thermophoresis (MST), cryogenic transmission electron microscopy (cryo-TEM), atomic force microscopy (AFM) and dynamic light scattering (DLS). We also produced and studied a CP mutant designed to slightly weaken the CP-CP interaction. Our results show that pH is a crucial factor influencing CP-CP interactions, but we also noticed the significant influence of ionic strength. We also observed that the physicochemical parameters of VLPs, such as diameter, polydispersity and thermal stability, are strongly influenced by the cargo. Interestingly, CP carrying mutations formed VLPs showing a wide range of possible diameters compared to the wild-type VLPs.

## 2. Results

### 2.1. Searching for Optimal Conditions of Empty VLP Formation

Earlier, it was shown that a decreasing pH is crucial for CP-CP interactions during VLP formation; however, the influence of ionic strength has not been taken into account. To assess the influence of both factors on the strength of the interaction between rCP and empty VLP assembly, we performed microscale thermophoresis (MST) assays. In the MST assay, a fluorescently labeled molecule (in our case rCP labelled with fluorescent marker) of constant concentration is titrated with a dilution series of the ligand (unlabeled rCP). The affinity between the proteins is assessed taking into account differences in a movement of the fluorescently labelled protein in a temperature gradient (thermophoresis) which is altered by any changes of the charge, size or hydration shell resulting from the ligand binding [23]. Two series of MST assays were performed: in Tris buffer at constant pH 7.5 with increasing salt concentration (150, 300, 600 and 1000 mM NaCl) and in Tris buffer at constant NaCl concentration of 150 mM and increasing pH (6.8, 7.5 and 8.5) (Figure 1A,B, respectively). The MST signals recorded for each measurement were plotted against rCP concentration to obtain dose-response curves, from which the K_d_ was evaluated. The rCP bound each other with a K_d_ in the nanomolar range at all tested conditions, which indicates a tight interaction. All the MST data collected clearly confirmed that both the pH and ionic strength significantly influence the interaction between rCPs. The K_d_ at pH 8.5 was 175 nM, that at pH 7.5 was 35 nM and that at pH 6.8 was 6.8 nM. At decreasing ionic strength, K_d_ reached 111 nM, 72 nM, 62 nM and 35 nM at 1000, 600, 300 and 150 mM NaCl, respectively. The influence of pH on the interaction between rCPs seems to be more significant than that of ionic strength, as illustrated by a two orders of magnitude higher range of K_d_ changes at pH gradient. 

To confirm the MST results, additional experiments were performed. The DLS analyses at the three mentioned pH points revealed that at pH 8.5 and 7.5, rCPs existed mostly in dimeric form (~6 nm hydrodynamic diameter) (Figure 1C). In addition, small fractions of particles with diameters corresponding to VLPs (~39–49 nm) and rCP aggregates (hydrodynamic diameter > 100 nm) were detected. At pH 6.8, particles of ~18 nm hydrodynamic diameter and a small population of particles with 286 nm diameter were observed. Interestingly, at pH 6.8, we did not detect rCP dimers. This result was verified by rCP cross-linking at pH 6.8 and 7.5 (Figure 1D). After the cross-linking reaction, the rCP complexes were analyzed by SDS PAGE. This confirmed that at pH 7.5, the main product was the rCP dimer, while at pH 6.8, rCP formed complexes/aggregates of relatively higher molecular weight than dimer. The results from DLS measurements and cross-linking reactions indicated that at pH 6.8, rCPs formed other kinds of rCP oligomers. To determine whether VLPs would be formed at pH 6.8, rCP was incubated in pH 6.8 buffer at 4 °C for 24 h. Subsequent DLS measurements revealed that VLPs did not form. Exclusively rCP oligomers 18 nm in diameter were present in the solution. In summary, the above observations indicated that a gradual decrease in both pH and ionic strength in the course of dialysis should enhance VLP formation. Thus, to obtain each type of VLP described in this study, we applied two-step dialysis.

### 2.2. Cored VLP Assembly

Under native conditions, the formation of the BMV capsid is initiated by interactions between CP and viral genomic RNA. This process is facilitated by the electrostatic interactions between the positively charged N-termini of CP and the negatively charged TLS located at the 3′ end of BMV genomic RNA [5,14,22]. Therefore, in the first experiments on VLP assembly, two-step dialyses were carried out in buffer containing yeast tRNA as a potential VLP assembly initiator. In the first step, the ionic strength and in the second step, the pH gradually decreased. A series of two-step dialyses were performed in which the tRNA:rCP molecular ratio was changed from 1:1 to 1:6. Prior to two-step dialyses, rCP was stored in buffer with high ionic strength and pH 7.2. After the second dialysis, rCP samples were subjected to native agarose gel electrophoresis. The obtained results indicated that tRNA stimulated the formation of VLPs (Figure 2). For the samples with tRNA:rCP ratios of 1:1 and 1:2, the band corresponding to native BMV was not observed. However, for both samples, in addition to the thick band at the bottom, which contained unbound tRNA, we observed additional thin bands in the middle of the lane, which might correspond to RNA-rCP complexes. For the sample with a tRNA:rCP ratio of 1:3, there was a thin band corresponding to the fully assembled VLPs and still many unbound tRNAs at the bottom of the lane. For samples with tRNA:rCP ratios of 1:4 to 1:6, only the bands migrating to a similar position as that of BMV were visible. Considering the above results in our further experiments on VLP assembly, a 1:6 tRNA:rCP mass ratio was applied. The fact that putative VLPs formed by rCP and tRNA moved slower in the agarose gel than native BMV might be explained by the differences in total charge and conformation between the two particles.

To determine whether polyelectrolytes other than RNA can be applied to assemble VLPs, we used 75-kDa polystyrene sulfonic acid (PSS), a synthetic polymer with a negative charge. PSS is capable of interacting with rCP and thus to some extent can mimic RNA, although PSS lacks complex secondary and tertiary structures. VLP assembly was analyzed in a series of two-step dialyses in which PSS:rCP mass ratios were changed from 1:1 to 1:8 (Figure 3) [24]. Agarose gel electrophoresis revealed that VLPs were formed exclusively in the reactions with the highest excess of CP, i.e., in the reaction with PSS-rCP mass ratios of 1:8. For subsequent experiments with PSS, a mass ratio of 1:8 (PSS:rCP) was used.

### 2.3. Properties of Empty and Cored VLPs

Three types of VLPs, empty VLPs (eVLPs), those with yeast tRNA (tVLPs) and those with polystyrene sulfonic acid (PVLPs), were characterized using a wide range of physicochemical methods. Cryo-TEM images revealed spherical structures of eVLPs and tVLPs (Figure 4A,C). The size of the former ranged from 20 to 65 nm with dominant fractions of 30–40 nm, and that of the latter ranged from 25 to 55 nm with dominant fractions of 30–40 nm (Figure 5A,C). In addition, cryo-TEM images confirmed that CP-CP interactions alone are strong enough to form spherical VLPs. Unfortunately, for PVLPs, we were unable to obtain cryo-TEM images.

The empty VLP analyses also involved atomic force microscopy (AFM). Experiments performed in air revealed that empty VLPs stably retained their spherical structure and dimensions in a dry environment (Figure 6). The average diameter of VLPs was 32 nm and ranged from 16 to 63 nm.

Consistent with the cryo-TEM results, DLS measurements also showed variations in hydrodynamic diameters both within and among the studied VLP populations (Figure 7). The eVLP population had the largest and most variable diameter, approximately 40 ± 10 nm. The diameter of tVLPs was approximately 28 nm ± 5 nm, and that of PVLPs was approximately 21 nm ± 5 nm.

To determine the thermal stability of VLPs, we used DLS to measure how their size changed with increasing temperature from 30 to 52 °C (Figure 8). At 30–47 °C, the major fraction of eVLPs had diameters ranging from 25–66 nm, which was consistent with our earlier observations. However, in this temperature range, a small population of eVLPs with diameters > 100 nm were also observed. At 48 °C and higher temperatures, the population of particles with diameters > 100 nm became dominant. tVLP were the most stable. Throughout the whole temperature range (30–52 °C), the tVLP hydrodynamic diameter was 18–34 nm. PVLP sizes ranged from 16–38 nm until 40 °C. Above 40 °C, some larger particles were formed (<70 nm), and above 44 °C, aggregates > 100 nm appeared.

### 2.4. Rational Designing of BMV-Based VLPs

The production of rCP in the heterologous system creates the possibility of obtaining VLPs with altered properties. To this end, we used molecular dynamics methods to design rCP mutant with a modified CP-CP interaction pattern (Figure 9 and Figure 10). Based on in silico-generated data, we expected that the exchange of two hydrophobic amino acids into hydrophilic groups (L123D and F183T) would weaken CP-CP binding. The rCP mutant was produced, and VLPs (MVLPs) were assembled in the same way as described above for unmutated ones. 

The formation of empty VLPs and with tRNA, from mutated rCPs (eMVLPs and tMVLPs, respectively) was confirmed with cryo-TEM (Figure 4B,D). However, similarly as for wild-type rCP, we failed to record cryo-TEM images of PMVLPs. The cryo-TEM images showed only a few eMVLPs (Figure 4B). On the other hand, the number of tMVLPs was significantly higher (Figure 4D). The eMVLPs were spheres of highly variable size from 35 to 80 nm (Figure 5B). The tMVLP spheres were smaller and more homogenous. Their size oscillated between 15 and 35 nm (Figure 5D). Interestingly, cryo-TEM analysis showed that tMVLPs were smaller than tVLPs (25–55 nm), which was confirmed by DLS analysis.

DLS measurements of mutated VLPs confirmed a wide size range of eMVLPs (Figure 7B). Mutated VLPs carrying negatively charged cores, and the wild-type VLPs, had larger diameters if formed with tRNA (tMVLPs: 27 nm ± 5 nm) than with PSS (PMVLPs: 24 nm ± 3 nm).

MVLPs also showed different stabilities than unmutated VLPs. The size of the eMVLPs varied within the whole range of temperatures applied in our experiment (from 19 to 85 nm). In the eMVLP sample, the amount of large aggregates (>100 nm) was significantly increased at 51 °C. We also observed that tMVLPs were not as stable as their tVLP analogs. At 46 °C, the size of tMVLPs was approximately 27–31 nm, but at higher temperatures, larger aggregates appeared. At 49 °C, a majority of tMVLPs were >300 nm. PMVLPs were smaller than other MVLPs until the temperature reached 41 °C. From 42 °C, the average size of the PMVLPs increased. At 51 °C and above, the majority of PMVLPs were >100 nm.

## 3. Discussion

There are two main types of interactions important for BMV capsid formation: CP-CP and RNA-CP. The lack of empty capsids in BMV-infected plants indicates that in vivo, RNA-CP interactions are pivotal for the formation of viral particles [25]. However, CP-CP interactions are sufficient to form BMV-based VLPs in vitro [26]. Our analysis showed that strong CP-CP binding depends on pH. We noticed that at the lowest tested pH of 6.8, rCPs tended to form an oligomeric structure, most likely rCP trimers proposed to initiate the nucleation process leading to capsid formation [18] (Figure 1). This result is consistent with the observation of Chevreuil and coworkers that a decrease in pH leads to stronger interactions between CPs of CCMV and more compact VLP formation [24]. Additionally, van Eldijk and coworkers showed that changing the pH can control empty CCMV capsid formation [26]. Here, we showed that the ionic strength may also significantly influence CP-CP interactions, suggesting that ionic interactions might be involved in empty VLP assembly apart from hydrophobic interactions. We also observed that a rapid decrease in pH induces the transformation of CP dimers into larger CP oligomers and blocks VLP formation. Thus, we proposed that two-step dialysis gradually decreasing the ionic strength and pH is a good method to assemble both empty and cored VLPs. This protocol proves to be particularly efficient for the production of tRNA-cored VLPs. Other types of VLPs, including empty capsids, were formed less effectively (Figure 4). Moreover, experiments involving empty capsids showed that VLPs preserve their structure even after drying (Figure 6).

Considering that virion assembly is based upon CP-RNA binding under native conditions, Rao distinguished two types of CP-RNA interactions that impact this process [25]. The first type depends on a specific RNA structure and involves the TLS located at the 3′ end of BMV genomic RNAs [14]. It was shown that the encapsidation of BMV RNA lacking TLSs is not effective but can be restored after complementation with yeast tRNA or separate TLSs [8]. We show that yeast tRNA alone is sufficient to mediate the formation of stable and homogenous VLPs. The mass ratio of yeast tRNA to rCP was similar to those shown by Cadena-Nava et al., which indicates that the length of the RNA molecule is less important [27]. Our results also confirm that the structural features of the cargo, apart from its negative charge, are important factors influencing VLP formation and features. Accordingly, VLPs containing tRNA better resembled native BMV capsids in terms of their size than VLPs containing PSS [28,29].

The second type of CP-RNA interaction, crucial for capsid formation, is sequence-independent electrostatic attraction between negatively charged RNA and the positively charged CP N-terminus. The importance of those interactions was demonstrated by Cadena-Nava and coworkers, who showed that CCMV CPs were able to encapsidate longer RNA than viral genomes [27]. The importance of a negative charge was also shown by Cheveruil and coworkers by encapsidating PSS in a VLP composed of CCMV CPs [24]. Here, we confirmed the above observations by showing that PSS mediated the formation of BMV rCP-based VLPs. In addition, our data suggest that due to the electrostatic interaction between rCP and cargo, VLPs are more compact. Thus, tVLPs and PVLP were smaller than eVLPs. As expected, the dimensions of VLPs also depended on the cargo size; accordingly, PVLPs were smaller than tVLPs and BMV capsids. Although PSS particles are longer than tRNA and do not form complex spatial structures, due to the hydrophobic core, they adopt compact structures in water environments. Chevreuil and coworkers obtained similar results studying VLPs based on CCMV CP [24]. The authors also explain this effect by the hydrophobic properties of PSS, which is poorly soluble in water and exists in a collapsed form [24]. Moreover, they noted that linear, flexible PSS chains require less free energy to be encapsulated than particles with stiff conformations, such as RNA [24,30].

In contrast to tVLPs and PVLPs, empty VLPs that rely only on weak CP-CP interactions are more flexible and less stable than those relying on strong CP-RNA or CP-PSS interactions. Accordingly, eVLPs, both wild-type and mutated, have a broader range of diameters than the corresponding loaded VLPs (Figure 5 and Figure 7). However, our designed eMVLP exhibits considerably higher assembly potential. The introduced mutations change the hydrophobic interaction between amino acid residues of two CPs into hydrophilic ones, which results in VLPs showing a wider range of diameters (Figure 10). Higher amounts of formed eVLPs in comparison to eMVLPs also indicate that hydrophobic interactions between two CPs (L123 and F183) are significant for BMV capsid assembly. The addition of cargo strongly influences MVLP formation (Figure 8B). Mutated VLPs containing tRNA were similar in size to wild-type tVLPs. This suggests that CP interactions with the tRNA structure strongly compensate for the weaker CP-CP interactions among the mutated CPs and are crucial for the formation of VLPs resembling native capsids. This observation once again confirms that the spatial conformation of tRNA strongly enhances VLP assembly, which was also presented by Rao et al. [25]. On the other hand, MVLPc-containing PSSs are smaller than wild-type PSSs and are more homogenous. PSS, as mentioned above, can form a collapsed structure that, together with weaker CP-CP interactions and higher flexibility of the mutated VLP shell, might result in better adjustment of rCPs to its cargo and bring it closer to the VLP’s core.

The results showing the thermal stability of the produced VLP are consistent with the above observations. We found that TLS and the negative charge of the cargo stabilize VLPs. PSS and tRNA stabilized wild-type capsid VLPs up to 40 °C. However, above this temperature, PVLPs were unstable, unlike tVLPs. As concluded earlier, TLSs and electrostatic interactions with rCPs significantly stabilize VLPs better than negative charge alone (Figure 8A). Our results indicate that tRNA presence is not mandatory for VLP formation but is crucial for particle stabilization. Mutated VLPs had slightly different thermal stability than wild-type VLPs. eMVLPs showed a wider range of sizes at room temperature than eVLPs. Interestingly, the aggregation of eMVLPs started to appear at 51 °C, while wild-type VLPs began to aggregate at 45 °C. This could be the effect of the stronger hydrophilic interactions resulting from the substitutions of two nonpolar amino acids with polar amino acids in the mutated CP. However, explaining this observation demands further study. For tMVLPs, 49 °C seems to be the temperature at which large aggregates start to prevail. Thus, tMVLPs seem to be less stable than tVLPs, which also indicates the significance of lateral CP-CP interactions on capsid stability. Interestingly, better adjustment of the cargo and CP in MVLPs was also confirmed by their higher thermal stability (aggregation proceeds at 42–51 °C) than that of VLPs (aggregation proceeds at 40–44 °C).

In summary, we have shown that recombinant BMV CP produced in a bacterial expression system is capable of self-assembly mediated by both CP-CP and CP-cargo interactions. By encapsidating tRNA and PSS, we highlighted the influence of cargo features on the structure and stability of BMV-based VLPs. We noted the profound influence of TLSs on the features of VLPs, while the negative charge of the cargo and CP-CP interactions seems to play a secondary but also important role. Moreover, we demonstrated that using genetic engineering, one can obtain VLPs with altered properties in comparison to VLPs formed by wild-type CP. Thus, the usage of a bacterial expression system for BMV CP production provides potential for the rational design of VLPs and research on their properties.

## 4. Materials and Methods

### 4.1. Protein Production and Purification

DNA coding for brome mosaic virus CP was cloned into the pMCSG48 expression plasmid and expressed in Rosetta^TM^ 2(DE3)pLysS competent *E. coli* cells (Novagen, Merck KGaA, Darmstadt, Germany) according to the manufacturer’s instructions. Briefly, bacterial cells were transformed with plasmids encoding wt or mutated BMV CP by a heat shock method. Then, each transformant was used to inoculate 250 mL of LB medium containing 100 mg/L ampicillin and 34 mg/L chloramphenicol. Bacteria were grown at 37 °C with shaking. When the OD600 of the culture reached 0.5–0.8, IPTG was added to a final concentration of 0.5 mM. After induction, bacteria were cultured for 16 h at 20 °C. The cells were harvested and frozen on dry ice for storage at −20 °C. Approximately 5 g of cells was suspended in 40 mL of lysis buffer I [25 mM Tris pH 8, 0.5 M NaCl, 0.2 mg/mL lysozyme (BioShop, Ontario, Canada), 250 U benzonase nuclease (Novagen,)] and sonicated. Cell debris was removed by centrifugation, and the supernatant was mixed with the same amount of buffer II [25 mM Tris pH 8, 0.5 M NaCl, 20 mM imidazole (BioShop)]. To purify BMV CP, affinity chromatography was used. After protein binding with Ni-NTA resin, the column was washed with buffer II. Next, the protein was eluted from the column using an imidazole gradient: 200, 300 and 500 mM in buffer II. The eluted protein was dialyzed using buffer III (25 mM Tris pH 8, 0.5 M NaCl). To remove the N-terminal tag, 1 mg of TEV protease was added per 10 mg of eluted protein. After TEV digestion, the sample was once again applied to a Ni-NTA charged column to remove the tag and any undigested protein. The first flowthrough was collected, and the purified proteins were concentrated by ultrafiltration (Amicon Ultra-10,000 MWCO, Millipore, Burlington, MA, USA). The concentration of the protein was determined by measuring the absorbance at 280 nm (NanoPhotometer N60, Implen, Westlake Village, CA, USA). The purity of the protein samples was assessed by SDS-PAGE, and the monodisperse state of the protein solutions was confirmed by dynamic light scattering (DLS) measurements using a Zetasizer µV (Malvern Panalytical, Malvern, GB) with the application of 6 µL of sample in a quartz cuvette with a 1 cm path length (Hellma QS 105.231). The correct molecular masses of all recombinant proteins were confirmed with Ultraflextreme MALDI Tof/Tof MS (Bruker Daltonics, Billerica, MA USA).

Two mutations, L123D and F183T, were introduced into the DNA sequence coding for BMV CP through PCR with specific primers. The mutated BMV CP sequence was then cloned into the pMCSG48 expression plasmid and amplified in DH5α competent *E. coli* cells (Thermo Fisher Scientific, Waltham, MA, USA). The plasmids were isolated and verified for L123D and F183T presence via DNA sequencing. Mutated protein expression, isolation and purification were performed as described above for the wild-type BMV CP. Isolated and purified BMV rCP was dialyzed against protein storage buffer (PSB) [1 M NaCl (BioShop), 1 mM EDTA (Merck), 20 mM Tris (BioShop), pH 7.2].

### 4.2. VLP Assembly

rCP (stored in the PSB) solution was subjected to a two-step dialysis process with the application of SnakeSkin Dialysis Tubing with a 3500 MWCO (Thermo Fisher Scientific). First, dialysis was performed against VLP assemble buffer (1) [50 mM NaCl (BioShop), 10 mM KCl (BioShop), 5 mM MgCl_2_·6H_2_O (Merck), 50 mM Tris (BioShop), pH 7.2] for 16 h at 4 °C. Next, dialysis was performed against VLP assemble buffer (2) [25 mM NaCl (BioShop), 10 mM KCl (BioShop), 25 mM NaOAc (Merck), 5 mM MgCl_2_·6H_2_O (Merck), 50 mM Tris (BioShop), pH 7.8] for 16 h at 4 °C. Afterwards, the samples were centrifuged (10 min, 10,000 rcf, 4 °C), and the supernatant was collected for further examination.

Assembly reactions of the VLP with tRNA or PSS were performed in a mass ratio gradient from 1:1 to 1:6 for tRNA:rCP and 1:1 to 1:8 for PSS:rCP. Assembly reactions proceeded as a two-step dialysis, as described above. Afterwards, the samples were centrifuged (10 min, 10,000 rcf, 4 °C), and the supernatant was transferred for further examination. Agarose gel electrophoresis was conducted to visualize the results of the VLP-core assembly gradient. A 1% agarose gel was used and stained with Midori Green (ABO). 10 µL of tVLP sample with 2 µL of loading dye (EURX, Gdańsk, Poland) and 25 µL of PVLP sample with 5 µL of loading dye was loaded on the gel. Samples were electrophoresed for 1.5 h at 80 V in 0.5× TBE buffer (BioShop). Gel was depicted on the ChemicDoc XRS+ System (BioRad, Hercules, CA, USA).

### 4.3. Cryo-TEM Measurements

Specimens for cryo-TEM imaging were prepared using CryoPunge 3 System (Gatan, Pleasanton, CA, USA). At room temperature and 95% relative humidity in the chamber, 4 µL droplet of each sample was put on a lacey carbon coated copper grid (Lacey C only, Ted Pella Inc., Redding, CA, USA) which were previously hydrophilized for 1 min using ELMO glow discharge system (Cordouan Technologies, Bordeaux, France). Subsequently to blotting the excess liquid, specimens were instantaneously plunged into a liquid ethan and shock frozen to ca. –183.15 °C. Vitrified samples were then transferred to Gatan 626 cryo-holder (Gatan, Pleasanton, CA, USA) that maintained temperature below −178.15 °C during the imaging. Imaging was carried out using Jeol JEM1440 transmission electron microscopy (Jeol Ltd., Tokyo, Japan) at accelerating voltage of 120 kV.

### 4.4. AFM Measurements

The VLP solution was diluted to a low concentration of 5 μg/mL, applied to a mica sample disc and left to dry. The dried sample was loaded onto a Multimode 8 instrument (Bruker, Billerica, MA, USA), and AFM tapping mode with an SNL-10 probe (Bruker) was used to record the shape of the prepared eVLPs. The results of the measurement were analyzed with NanoScope Analysis 1.5 software.

### 4.5. DLS Measurements

DLS measurements were recorded using a Zetasizer uV instrument (Malvern Instruments). The measurements were performed at a wavelength λ = 488 nm and at a 90° incidence angle of the light beam; a 2 µL quartz cuvette was used. The concentration of rCP was 0.1 mg/mL. Twelve measurements were performed for each VLP at 25 °C for standard DLS measurements and at temperatures ranging from 30–52 °C to analyze the thermal stability of the VLPs.

### 4.6. MST Experiments

Purified rCP was transferred to MST assay buffer for ionic strength studies (50 mM Tris, pH 7.5: 5 mM MgCl_2_ and 150, 300, 600, or 1000 mM NaCl) or pH studies (50 mM Tris, pH 8.5, 7.5 or 6.8:5 mM MgCl_2_ and 150 mM NaCl) using Zeba™ Spin desalting columns (ThermoFisher Scientific). CP was fluorescently labeled following the manufacturer’s instructions (Nanotemper Technology, Munich, Germany), i.e., using Monolith NT^TM^ Protein Labeling Kit RED-NHS 647. The concentration of labeled proteins was adjusted to 50 nM. A dilution series (starting from 15 μM) of up to 16 unlabeled protein concentrations was prepared in a final volume of 10 μL. Ten microliters of labeled protein was added to each dilution and mixed with a pipette. All samples were centrifuged for 5 min at 10,000 *g* prior to MST measurement. Capillaries were filled and loaded into a Monolith NT.115 instrument, and the thermophoresis experiment was performed. The results were analyzed with MO.Affinity Analysis software (Nanotemper).

### 4.7. Protein Cross-Linking

CP crosslinking was performed using 5 mM disuccinimidyl suberate (DSS, Thermo Scientific) in DMSO according to the manufacturer’s instructions using rCP in MST buffer described above at pH 7.5 and 6.8. The reaction was stopped after 5, 10, 15 and 30 min by adding 1 M TRIS-HCl, pH 7.5. The crosslinking results were visualized by SDS-PAGE.

### 4.8. CP Mutant Design

The hexamer model of the BMV coat protein and of the mutant variant were constructed using the UCSF Chimera program [31] on the basis of the 1JS9 structure from the PDB repository. Proper protonation and parameterization of protein models was performed using the pdb2gmx module of the GROMACS 4.6.7 package [32], employing the Amber99SB-ILDN [33] force field. CP hexamers were immersed in a truncated dodecahedral simulation box containing TIP3P model water molecules. Solvated systems were subsequently neutralized, and an excess of Na^+^ and Cl^−^ ions was added to obtain a final concentration of 0.1 M NaCl.

The geometry of the systems was optimized using the energy minimization technique with the steepest descent algorithm. Next, optimized systems were relaxed during 200 ps NVT ensemble simulation runs and subsequently 200 ps NPT runs. Such relaxed systems were used as starting structures for final, 50 ns production NPT runs. During all runs, the time step was 2 fs. During equilibration runs, the output frequency was set to 100 time steps (200 fs), whereas in production simulations, it was set to 1000 time steps (2 ps). Long-range electrostatic interactions were calculated using the particle mesh Ewald method, with a cutoff value of 1 nm. Velocity rescaling was used to scale the temperature and Parinello-Rahman barostat to maintain constant pressure in the simulated systems.

## Figures and Tables

**Figure 1 ijms-22-03098-f001:**
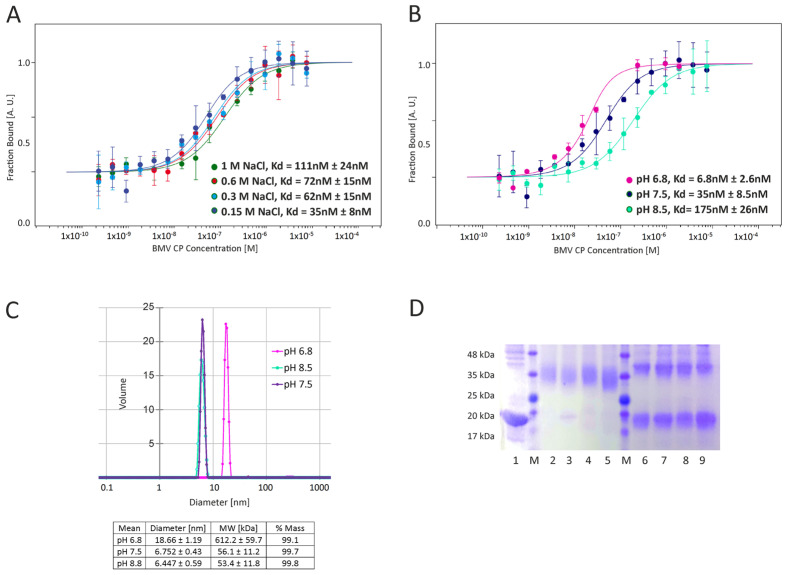
CP-CP affinity analysis in various buffer conditions. (**A**) MST results for the interaction in NaCl gradient. (**B**) MST results for the interaction in pH gradient. The binding curves and the corresponding K_d_ values resulted from the analysis of raw MST data. The curves and K_d_ values estimated from the same measurement are marked with the same colors. (**C**) DLS analysis of rCP hydrodynamic diameter at concentration of 0.1 mg/mL in a pH gradient (volume distribution). The colors of DLS curves correspond to the results from (**B**). (**D**) The cross-linked CP visualized via SDS-PAGE. Lanes: 1—CP without cross-linking, M—molecular weight marker, 2–5—CP crosslinked at pH 7.5 after 5, 10, 15 and 30 min, respectively, 6–9—CP crosslinked at pH 6.8 after 5, 10, 15 and 30 min, respectively.

**Figure 2 ijms-22-03098-f002:**
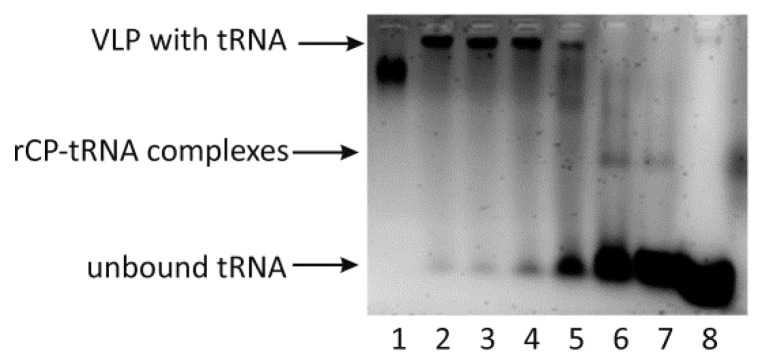
BMV-based VLPs formation according to various CP/tRNA mass ratios in native agarose gel electrophoresis. Lanes: 1—wt BMV, 2–7—various CP/tRNA mass ratios (from 6:1 to 1:1), 8—tRNA without CP. The arrows indicate formed VLP with tRNA inside, intermediate CP-tRNA complexes and the unbound tRNA.

**Figure 3 ijms-22-03098-f003:**
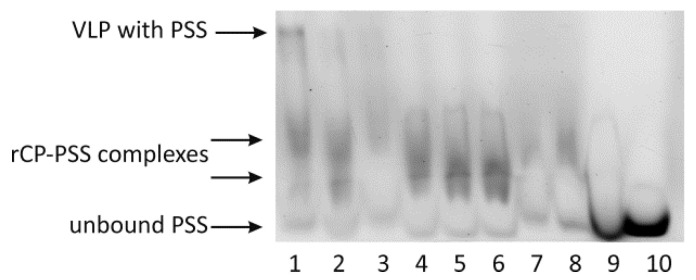
BMV-based VLPs formation according to various CP/PSS mass ratios in native agarose gel electrophoresis. Lanes: 1 to 8—various PSS/CP mass ratios (from 1:8 to 1:1). Lane 9—PSS without protein. Lane 10—tRNA without protein. The arrows indicate formed VLP with PSS inside, intermediate CP-PSS complexes and the unbound PSS.

**Figure 4 ijms-22-03098-f004:**
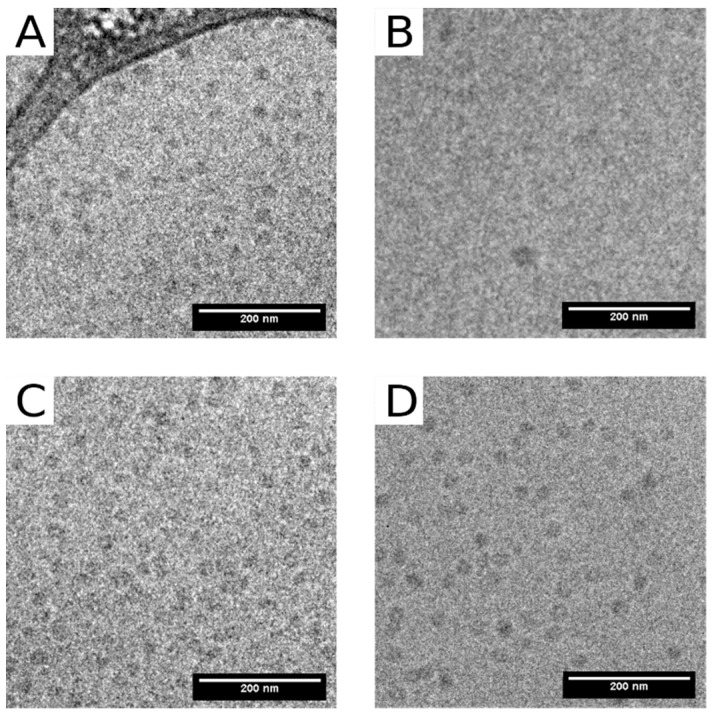
Cryo-TEM images of BMV-based VLPs. (**A**) Empty VLPs obtained from wt CP (eVLPs). (**B**) Empty VLPs obtained from mutated CP (eMVLPs). (**C**) VLPs obtained from wt CP with tRNA (tVLPs). (**D**) VLPs obtained from mutated CP with tRNA (tMVLPs).

**Figure 5 ijms-22-03098-f005:**
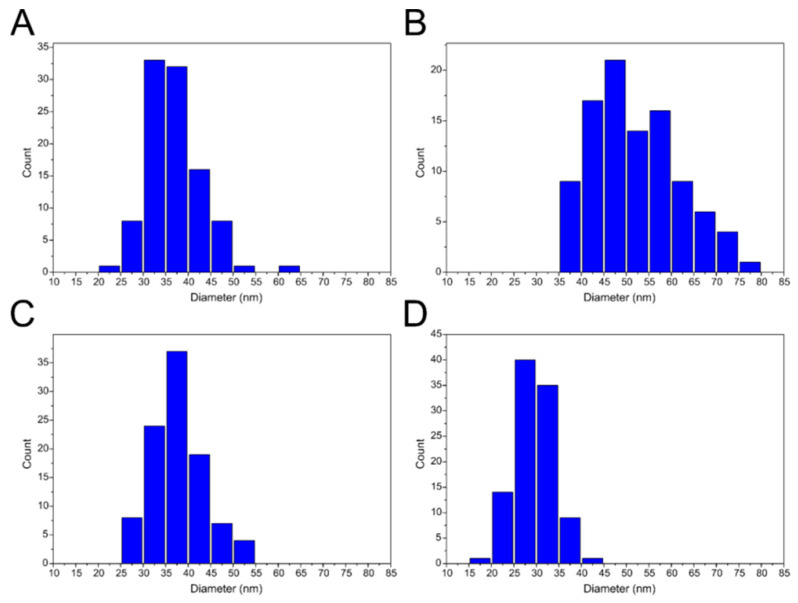
Size distribution of VLP populations from cryo-TEM images. (**A**) Empty VLPs obtained from wt CP (eVLPs). (**B**) Empty VLPs obtained from mutated CP (eMVLPs). (**C**) VLPs obtained from wt CP with tRNA (tVLPs). (**D**) VLPs obtained from mutated CP with tRNA (tMVLPs).

**Figure 6 ijms-22-03098-f006:**
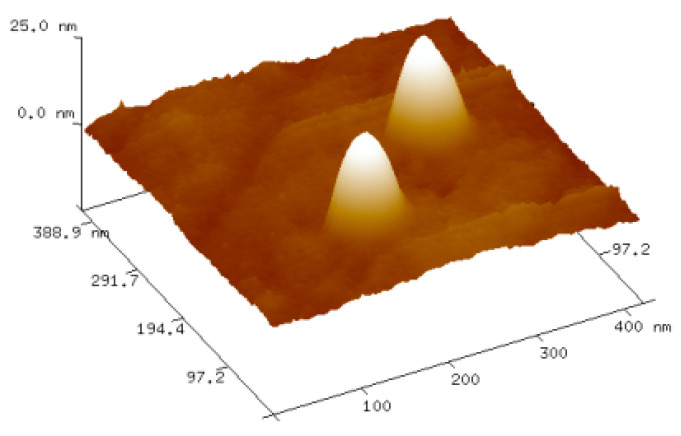
Exemplary picture of empty VLPs recorded using AFM tapping mode.

**Figure 7 ijms-22-03098-f007:**
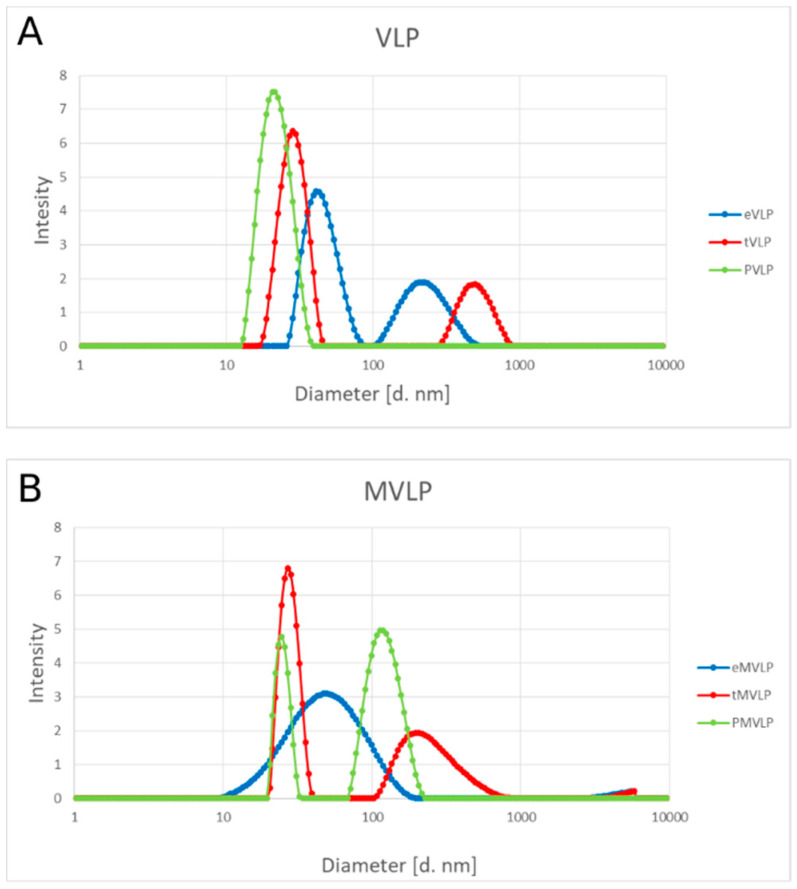
Size distribution of BMV-based VLPs in sodium acetate buffer (pH 4.8) and 25 °C by dynamic light scattering (DLS). Wild-type VLPs (**A**) and mutated VLPs (**B**). Empty VLPs are in blue, VLPs with tRNA are in red, and VLPs with PSS are in green.

**Figure 8 ijms-22-03098-f008:**
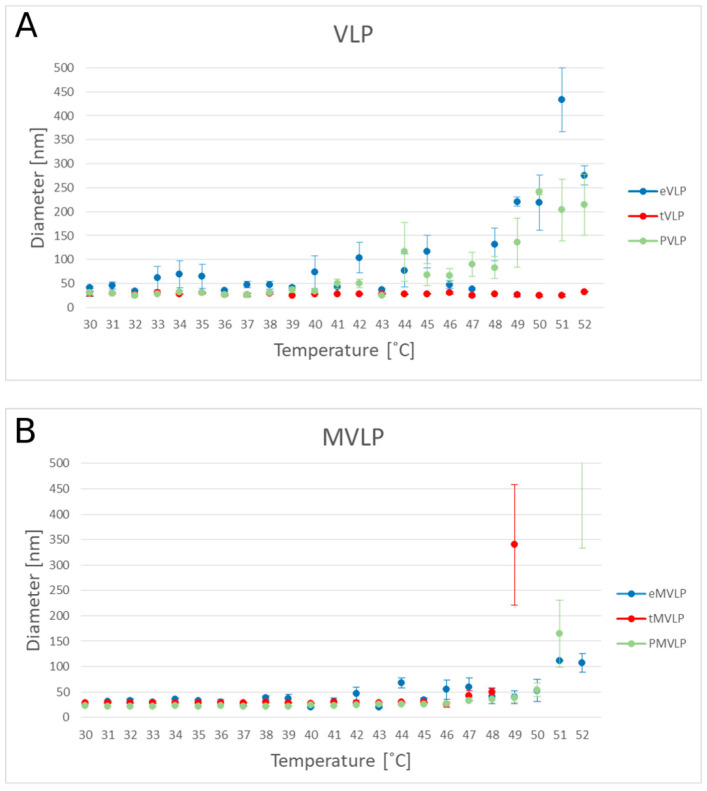
BMV-based VLPs size as a function of temperature. The diameter of VLPs was measured by DLS through a gradient of increasing temperature (30–52 °C). Wild type VLPs (**A**) and mutated VLPs (**B**). Empty VLPs (eVLPs (**A**) and eMVLPs (**B**)) are in blue, VLPs with tRNA (tVLPs (**A**) and tMVLPs (**B**)) are in red, and VLPs with PSS (PVLPs (**A**) and PMVLPs (**B**)) are in green.

**Figure 9 ijms-22-03098-f009:**
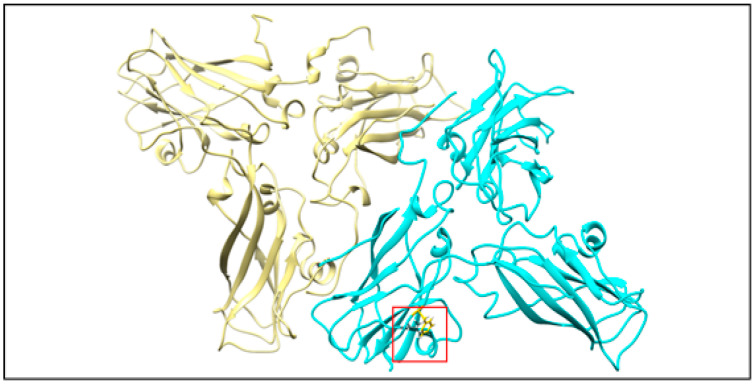
Computational model of two wild-type BMV capsid protein trimers. One trimer in yellow, second in cyan. The red square shows the place of interaction between two amino acid residues on two neighboring chains—the target for our mutation.

**Figure 10 ijms-22-03098-f010:**
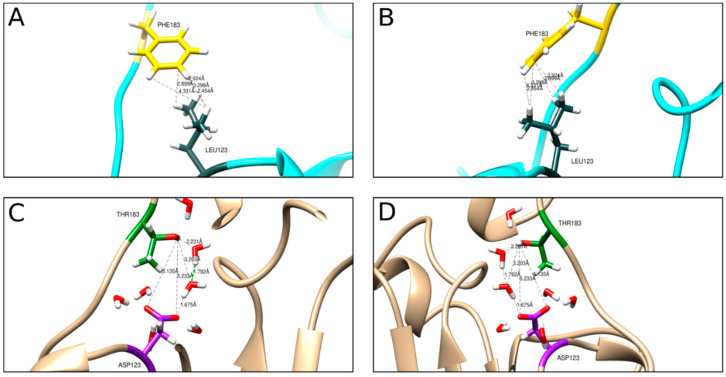
Computational model of a fragment of the BMV capsid protein. (**A**,**B**) depict wild-type virus hydrophobic interactions of two interacting amino acid residues on two neighboring chains. Leu123 in slate gray, Phe183 in yellow, rest of the protein in cyan. Black dashed lines show distances (in Å) between residues involved in hydrophobic contacts. (**C**,**D**) depict the hydrogen bond interaction pattern between mutated residues. Asp123 is shown in magenta, Thr183 in green, and the rest of the protein in beige. The electrostatic interactions between residues are mediated by water molecules. Black dashed lines show electrostatic interactions between residues and residue/water molecules.

## Data Availability

The data presented in this study are available on request from the corresponding author.

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
