# Peer review of "Virus-Like Particles Produced Using the Brome Mosaic Virus Recombinant Capsid Protein Expressed in a Bacterial System"

_ijms, 2021, doi:10.3390/ijms22063098_

Round 1
Reviewer 1 Report
This manuscript describes detailed analyses on the in vitro self-assembling behavior of recombinant BMV capsid by electrophoresis, microscale thermophoresis, electron microscopy and DLS. The manuscript is well-written for broad readers and the perspective describes the important subjects to be achieved in this area. Therefore, this manuscript would be suitable for the publication on IJMS after minor modification for a following comment.
- In the first paragraph of the Introduction, the authors should cite several important papers or reviews on application of viral capsids to nanotechnology and biomedical field. For example, Chem. Soc. Rev., 2016, 45, 4074; Adv. Healthcare Mater., 2016, 5, 1386; Chem. Commun., 2018, 54, 8944; Small 2011, 7, 1609; Acc. Chem. Res. 2011, 44, 774; Org. Biomol. Chem. 2007, 5, 2891; WIREs Nanomed. Nnobiotech., 2016, 8, 554.
- In addition, the authors should cite reviews about recent progress on physicochemical analyses of self-assembly of viral capsids in the first paragraph of the Introduction. For example, Nature Review Physics, 2021, 3, 76.
- I think many readers (including me) would not familiar to microscale thermophoresis (MST) assays. Please explain the principle of MST assay briefly in section 2.1.
- It seems that the scale on the horizontal axis (1, 10, 100, 1000) in Fig. 1C (DLS) is slightly off. In addition, the CP concentration in DLS measurements should be shown in the caption of Fig.1.
Author Response
1. In the first paragraph of the Introduction, the authors should cite several important papers or reviews on application of viral capsids to nanotechnology and biomedical field. For example, Chem. Soc. Rev., 2016, 45, 4074; Adv. Healthcare Mater., 2016, 5, 1386; Chem. Commun., 2018, 54, 8944; Small 2011, 7, 1609; Acc. Chem. Res. 2011, 44, 774; Org. Biomol. Chem. 2007, 5, 2891; WIREs Nanomed. Nnobiotech., 2016, 8, 554.
Ad 1. Appropriate sentence and the citiations in the introduction have been added.
2. In addition, the authors should cite reviews about recent progress on physicochemical analyses of self-assembly of viral capsids in the first paragraph of the Introduction. For example, Nature Review Physics, 2021, 3, 76.
Ad 2. Appropriate sentences and the citiations in the introduction have been added.
3. I think many readers (including me) would not familiar to microscale thermophoresis (MST) assays. Please explain the principle of MST assay briefly in section 2.1.
Ad 3. Explanation of the MST principle has been added at the beginning of 2.1 section.
4. It seems that the scale on the horizontal axis (1, 10, 100, 1000) in Fig. 1C (DLS) is slightly off. In addition, the CP concentration in DLS measurements should be shown in the caption of Fig.1.
Ad 4. Fig 1C has been improved and the CP concentration has been added to the caption of Fig.1.
Reviewer 2 Report
In this manuscript, the authors used the recombinant capsid protein of brome mosaic virus (BMV) to study the effect of ionic strength, pH, and the encapsulated cargo on the assembly of virus-like particles (VLPs). They found that pH and ionic strength significantly affected CP-CP interactions, and the cargo played an important role in determining structural features and stability of BMV-based VLPs. Using a protocol of two-step dialyses, they produced different VLPs and characterized those empty and cored VLPs with different methods. The experimental data were carefully presented, interpreted, and discussed in the context of related literatures. The results are meaningful and provides important insight into the factors that influence the assembly of BMV-based VLPs. One specific comment is that the author should describe their major findings (not just described what they did) in Abstract.
Author Response
1. One specific comment is that the author should describe their major findings (not just described what they did) in Abstract.
Ad 1. The Abstract has been improved according to the Reviewer's suggestions.
The improved parts have been marked yellow in the new version below:
Virus-like particles (VLPs), due to their nanoscale dimensions, presence of interior cavities, self-organization abilities and responsiveness to environmental changes, are of interest in the field of nanotechnology. Nevertheless, comprehensive knowledge of VLP self-assembly principles is incomplete. VLP formation is governed by two types of interactions: protein – cargo and protein – protein. These interactions can be modulated by the physicochemical properties of the surroundings. Here, we used brome mosaic virus (BMV) capsid protein produced in an E. coli expression system to study the impact of ionic strength, pH and encapsulated cargo on the assembly of VLPs and their features. We showed that empty VLP assembly strongly depends on pH whereas ionic strength of the buffer plays secondary but significant role. Comparison of VLPs containing tRNA and polystyrene sulfonic acid (PSS) revealed that the structured tRNA profoundly increases VLPs stability. We also designed and produced mutated BMV capsid proteins that formed VLPs showing altered diameters and stability compared to VLPs composed of unmodified proteins. We also observed that VLPs containing unstructured polyelectrolyte (PSS) adopt compact but not necessarily more stable structures. Thus, our methodology of VLP production allows for obtaining different VLP variants and their adjustment to the incorporated cargo.
Reviewer 3 Report
The authors varied a condition of BMV VLPs formation and investigated an effect of ion concentration and pH. Mutated CPs with hydrophilic CP-CP interaction were designed and produced to modify a property of VLPs. The mutated CPs formed capsids as similar as a wild type did. However, the interaction property with a load of VLP were modified. It is interesting that the mutated CPs formed VLPs even when hydrophobic CP-CP interaction were replaced to hydrophilic one.
Fig. 1C The peak value of the curve looks bigger than the value indicated in the table. Why do these values not match?
Fig. 4, 5, 8 An explanation and a mention about tMVLPs sample is lacking in the main text. The description should be put around the figures.
Fig. 8 Is the fitting curves rational and suitable for this experiment? If so, the authors should clarify the formula and the reason.
Fig. 9 I can not see “the red square” in the figure.
Fig. 10 The dashed lines looks not “yellow”.
Author Response
1.Fig. 1C The peak value of the curve looks bigger than the value indicated in the table. Why do these values not match?
Ad 1. Fig 1C has been improved. The discrepancy between the hydrodynamic diameter values in the graph and in the table had resulted from an error which occured during the preparation of the Fig 1C.
2. Fig. 4, 5, 8 An explanation and a mention about tMVLPs sample is lacking in the main text. The description should be put around the figures.
Ad 2. Appropriate explanations have been added to the Fig 4, 5 and 8 ant their captions, as suggested by the Reviewer.
3. Fig. 8 Is the fitting curves rational and suitable for this experiment? If so, the authors should clarify the formula and the reason.
Ad 3. The fitting curves have been removed from the graphs at Fig. 8.
4. Fig. 9 I can not see “the red square” in the figure.
Ad 4. Red square in Fig. 9 havs been enlarged.
5. Fig. 10 The dashed lines looks not “yellow”.
Ad. 5. The caption of Fig.10 has been improved.